

# Anatomy and behavior of *Laternula elliptica*, a keystone species of the Antarctic benthos (Bivalvia: Anomalodesmata: Laternulidae)

Flávio Dias Passos[1], André Fernando Sartori[2,3], Osmar Domaneschi[3] and Rüdiger Bieler[4]

[1] Department of Animal Biology, Institute of Biology, University of Campinas (UNICAMP), Campinas, São Paulo, Brazil
[2] THIS Institute, University of Cambridge, Cambridge, United Kingdom
[3] Department of Zoology, Institute of Biosciences, Universidade de São Paulo, São Paulo, Brazil
[4] Negaunee Integrative Research Center, Field Museum of Natural History, Chicago, Illinois, United States

Corresponding authors
Flávio Dias Passos,
flaviodp@unicamp.br
Rüdiger Bieler,
rbieler@fieldmuseum.org

## ABSTRACT

*Laternula elliptica* (P. P. King, 1832) is the sole representative of the anomalodesmatan family Laternulidae and the largest bivalve in the Antarctic and Subantarctic. A keystone species of the regional benthic communities, it has reached model status, having been studied in hundreds of scientific works across many biological disciplines. In contrast, its anatomy has remained poorly known, with prior published data limited to partial descriptions based on chemically preserved specimens. Based on observations of aquarium-maintained living animals at the Brazilian Comandante Ferraz Antarctic Station, gross-morphological dissections, and histological sectioning, the comparative anatomy, functional morphology, and aspects of behavior of *L. elliptica* are described and discussed. Special focus is placed on the pallial organs (including elucidation of cleansing and feeding sorting mechanisms in the mantle cavity) and the musculature. Among the noteworthy findings are the presence of well-developed siphons furnished with sensory tentacles at its tips, some of which bearing eyes; large, folded gills and labial palps capable of sorting the material entering the mantle cavity; an inter-chamber communication in the posterior region of the mantle cavity; an ample ventral mantle fusion with an anterior pedal gape; the absence of a 4th pallial opening; and the absence of a ligamental lithodesma in adult specimens. This study reevaluates the available anatomical data in the literature, both supplementing and correcting previously published accounts.

## INTRODUCTION

*Laternula elliptica* (P. P. King, 1832), the sole representative of the Laternulidae in Antarctic and subantarctic waters, is ubiquitous along its circumpolar distribution and is also known from the South Shetland, South Orkney, South Sandwich, South Georgia and

Kerguelen Islands (*Soot-Ryen, 1951*; *Dell, 1990*). The species, which is known from the region since the Pliocene (*Linse et al., 2006*), is considered a sister taxon to other extant species of *Laternula* (*sensu lato*) from Australia and the central Indo-West Pacific, with the species-level diversity of temperate and tropical members of the genus in need of investigation (*Taylor et al., 2018*; *MolluscaBase, 2022*).

The soft-substratum species has been collected from the intertidal to continental slope depth of about 700 m (*Waller et al., 2016*), but with almost all live-collected records from depths shallower than 100 m (*Dell, 1990*; *Engl, 2012*). *Nicol (1966)*, *Morton (1976)*, and *Narchi, Domaneschi & Passos (2002)* described the shell valves in detail (shown in Fig. 1). Compared to its lower latitude relatives of the family, *L. elliptica* is larger and thicker-shelled (*Watson et al., 2012*; *Prezant, Shell & Wu, 2015*) and lacks the spinules on the shell surface recorded from other species (*Checa & Harper, 2010*). *L. elliptica* is a simultaneous hermaphrodite, producing large eggs (about 200 µm in diameter), which develop as encapsulated lecithotrophic larvae (*e.g.*, *Ansell & Harvey, 1997*; *Kang, Ahn & Choi, 2003*).

*Smith (1902*: 210) already highlighted this species as "the giant of its genus" *Anatina* (then encompassing what is now the family Laternulidae). As the largest (>100 mm shell length) and very abundant bivalve (*e.g.*, 87 ind.m-2 in Collins Harbour, King George Island; *Ahn, 1994*), it dominates benthic communities (*Stout & Shabica, 1970*; *Hardy, 1972*; *Momo et al., 2002*; *Urban & Mercuri, 1998*; *Zamorano, Duarte & Moreno, 1986*), and is considered a keystone species of the Antarctic benthos (*Harper et al., 2012*). Its wide distribution in the Antarctic realm, high abundance, ease of collection, and ability to survive under experimental conditions have allowed it to reach model status, having been studied in hundreds of scientific articles (*Waller et al., 2016*) representing a broad spectrum of biological disciplines. Among these are investigations focusing on metabolism and energy budget (*e.g.*, *Agüera et al., 2017*; *Ahn & Shim, 1998*; *Momo et al., 2002*), biochemistry (*Ahn, 2000*; *González & Puntarulo, 2011*), heavy metal concentrations and pollution (*Ahn et al., 1996*; *Lister, Lamare & Burritt, 2015*; *Wing et al., 2020*), shell composition and structure (*Barrera et al., 1994*; *Nehrke et al., 2012*; *Sato-Okoshi & Okoshi, 2008*), reproduction and larval development (*Ansell & Harvey, 1997*; *Bigatti, Penchaszadeh & Mercuri, 2001*; *Kang, Ahn & Choi, 2003*, *2008*; *Pearse, Bosch & McClintock, 1986*; *Pearse et al., 1987*; *Powell, Tyler & Peck, 2001*), ageing (*Peck, Powell & Tyler, 2006*; *Philipp, Pörtner & Abele, 2005*), ocean acidification and warming (*Bylenga, Cummings & Ryan, 2015*, *2017*; *Cummings et al., 2011*), thermal stress and hypoxia (*Kim et al., 2009*; *Morley et al., 2007a*, *2009a*, *2009b*, *2012*; *Park et al., 2008*; *Peck, Pörtner & Hardewig, 2002*; *Peck et al., 2004*; *Pörtner, Peck & Hirse, 2006*), and iceberg scouring (*Harper et al., 2012*; *Philipp, Husmann & Abele, 2011*). Numerous molecular studies have been applied to the species, from assembling the complete mitochondrial genome (*Park & Ahn, 2015*), transcriptomics (*Clark et al., 2010*), and studying heat shock proteins (*Ramsøe, Clark & Sleight, 2020*; *Truebano et al., 2013*), to treating it as the exemplar for its family in class-wide phylogenetic studies (*Bieler et al., 2014a*, *2014b*; *Combosch et al., 2017*).

However, none of the many published studies focusing on this otherwise well-known species has ever dealt in-depth with its anatomy. For a long time, anatomical knowledge

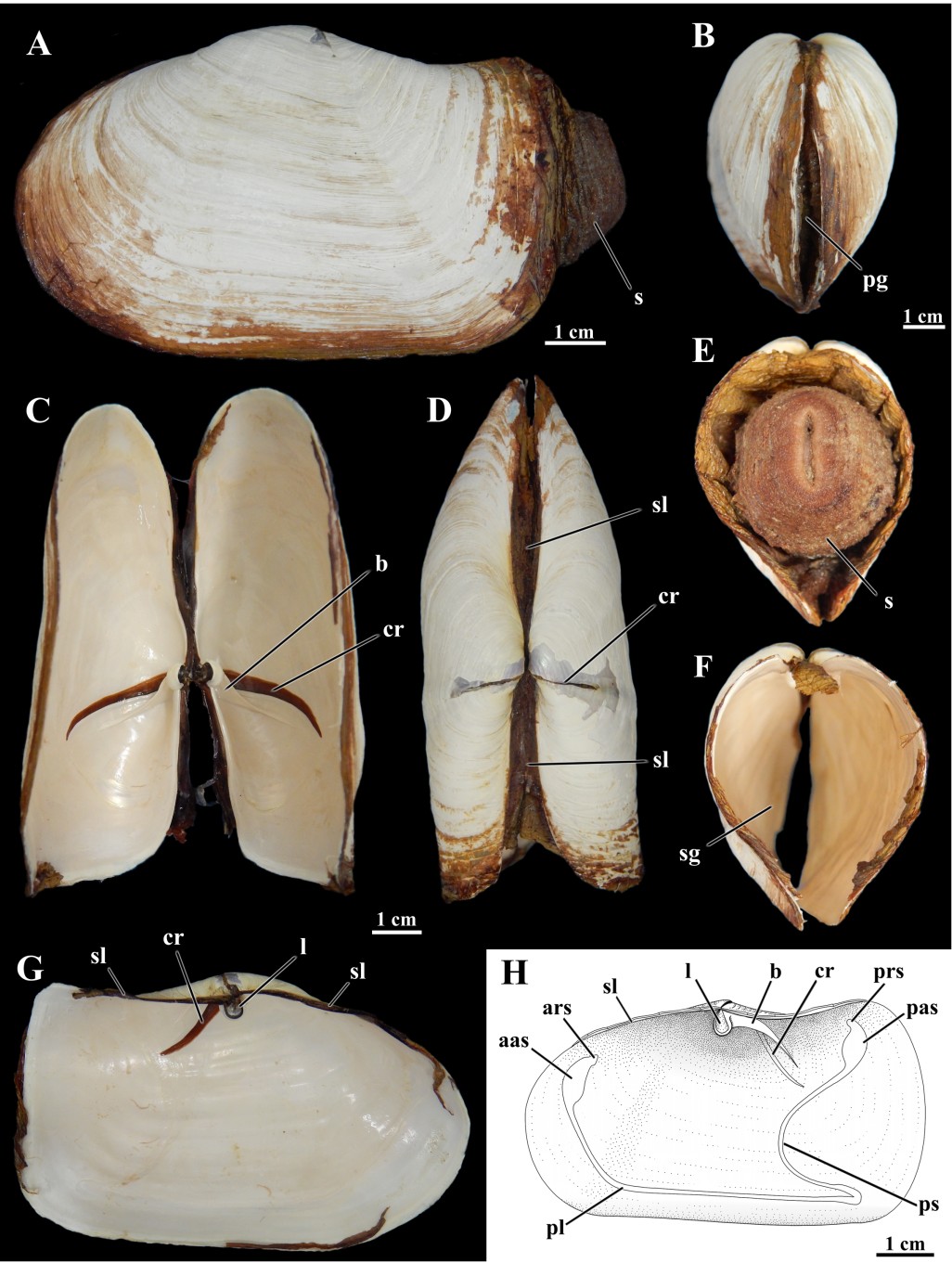

**Figure 1 Shell of *Laternula elliptica*.** (A–G) From the same specimen (ZUEC BIV 8397): (A) Outer left view. (B) Anterior view. (C) Ventral view with valves partially opened. (D) Dorsal view. (E) Posterior view, with preserved soft parts. (F) Same, without soft parts. (G) Inner view of the left valve. (H) Sketch of inner surface of a right valve. B, E, F and C, D, G are at the same scales, respectively. Abbreviations: aas, anterior adductor muscle scar; ars, anterior pedal retractor muscle scar; b, buttress; cr, crack filled with periostracum; l, ligament attached to the chondrophore; pas, posterior adductor muscle scar; pg, pedal gape; pl, pallial line; prs, posterior pedal retractor muscle scar; ps, pallial sinus; s, siphons; sg, siphonal gape; sl, secondary ligament.

has remained limited to the work of *Burne (1920)*, who provided an incomplete description based on a damaged individual specimen. During the austral summers of 1996–1997 and 1997–1998, Professor Osmar Domaneschi had the opportunity to conduct aquarium-assisted observations of living animals over several weeks during research visits to the Brazilian Comandante Ferraz Antarctic Station, resulting in detailed drawings and associated notes toward a planned manuscript. Unfortunately, the research remained unpublished. The most comprehensive published treatment of *L. elliptica* appeared in the work by *Bieler et al. (2014a, 2014b)*. Unaware of Domaneschi's field studies of living animals, Bieler et al. based their data on the analysis of preserved material (FMNH BivAToL-202), originally collected at the British Antarctic Survey's Rothera Research Station, Adelaide Island, Antarctic Peninsula. Other morpho-anatomical data were provided by *Peck et al. (2004)* on the anatomy of the organs concerned in the burrowing and surface movements and by *Sartori, Passos & Domaneschi (2006)* on the occurrence of arenophilic glands in both the mantle edge and surrounding the siphonal openings.

Before his untimely death in 2008, Domaneschi had entrusted his students (F.D.P. and A.S.) with his drawings and notes. The current publication utilizes many of the original illustrations and observations from that material. This article reviews the comparative anatomy, functional morphology, and aspects of behavior of *L. elliptica*, with special focus on the pallial organs and musculature. Based on original information from living specimens, this study reevaluates literature data, both supplementing and correcting previously published accounts.

## MATERIALS AND METHODS

In the austral summers of 1996–1997 and 1997–1998, living specimens of *Laternula elliptica* were collected from muddy and muddy-sand substrata at depths of 5 to 20 m in the Admiralty Bay, King George Island, Antarctica (62°05′S–58°23′W), both using a Van Veen grab and manually by SCUBA divers. Many living and intact specimens removed from undisturbed bottoms, as well as severely damaged specimens found unburied along new iceberg scours, were kept in aquaria with natural sediment and 33%, circulating seawater at $0 \pm 1\,°C$ at the Brazilian Comandante Ferraz Antarctic Station (EACF) on King George Island. In 1996–1997, twenty whole specimens with shell length ranging from 1.0 to 4.0 cm ($n = 10$) and 5.0 to 9.6 cm ($n = 10$) were allowed to bury in isolated aquaria, each containing *circa* 13 cm depth of natural muddy sediment, and their surface movements recorded over a four-week period. The morpho-functional analysis began at that time and continued in 1997–1998, through observations of both living and preserved specimens dissected under a stereomicroscope. In dissected animals, with one of its valves removed, cleansing, feeding, and sorting mechanisms in the mantle cavity were elucidated using powdered carmine, graded mineral grains, and natural fine organic particles, which were precipitated over their epithelia.

After finding a wide opening between the supra- and infra-branchial chambers in the first dissected specimens, every specimen was checked to confirm the presence/absence of such an opening. To ascertain that the opening was not an artifact of dissection, seven living, intact specimens (1.0 through 8.0 cm in shell length) were tested on their ability to

quickly transfer water from the exhalant into the inhalant chamber. These specimens had the exhalant siphon lumen injected with a highly concentrated carmine suspension and were immediately stimulated by forceps both to contract and tightly close the exhalant opening. One living, minute (1.0 mm in shell length) specimen was prepared for SEM analysis using the same methods applied in previous studies of other Antarctic bivalve species (*Passos, Domaneschi & Sartori, 2005*; *Passos & Domaneschi, 2006*; *Passos, Meserani & Gros, 2007*; *Passos & Domaneschi, 2009*); its shell valves and mantle lobes were excised to observe this passage between the two chambers through a higher magnification.

For routine serial sectioning, a complete 1.7 cm specimen and excised organs of larger specimens from the original (1996–1998) collecting events were fixed in Bouin's fluid, embedded in paraffin, and sectioned at 7 μm. These histological sections from this older material were, however, not in excellent condition. In 2001, the first author of the present article had the opportunity to collect and dissect some fresh specimens at the same site, from which portions of the ctenidia and optic tentacles were prepared by embedding the tissue in glycol methacrylate Leica Historesin and sectioning transversely and sagittally at 3 μm, following the methodology described by *Passos, Domaneschi & Sartori (2005)*. All histological sections were stained with haematoxylin and counterstained with eosin.

Voucher specimens of this study are deposited in the molluscan collection of the Museum of Zoology, UNICAMP, numbers ZUEC BIV 7570–7633, 8374–8390, and 8397–8399.

## RESULTS

### Shell

The shell of *L. elliptica* from the Admiralty Bay population (Fig. 1) matches the general characterization given by *Nicol (1966)*, *Morton (1976)*, and *Narchi, Domaneschi & Passos (2002)*.

Shells in the material examined ($n$ = 40) varied from 1.0 to 9.7 cm in length; some specimens exhibited evidence of injury in one or both shell valves, followed by regeneration of the nacreous layers only. The brownish periostracum was usually masked by loosely adhered particles from the surrounding sediment; particles attached to the shell surface by arenophilic threads as described for related species (*Sartori, Passos & Domaneschi, 2006*) were not present. The valves are connected by an edentulous hinge, where there is a robust internal ligament attached to chondrophores (Fig. 1); a lithodesma was not observed in the material examined. However, because hinge structure was not analyzed in every available specimen, it is possible that a lithodesma is present in specimens less than 1.5 cm in shell length, as reported by *Sartori (2009)* in specimens from Hangar Cove, Adelaide Island. Knife-like calcareous ridges support the chondrophores, functioning as strengthening buttresses or clavicles, and extend postero-ventrally from each of the valves' umbonal cavities; nearly anterior and parallel to each of these buttresses there is a long, periostracum-filled fissure (= dorsal crack) in the umbonal and disk regions visible from both the internal and external surfaces of the valves. The small, elliptical-elongated anterior and posterior adductor muscle scars are fused to the dorsally placed anterior and posterior pedal retractors scars, respectively; right and left pedal

protractor muscle scars are ventrally fused to the anterior adductor scar. The well-marked, entire pallial line is slightly distanced from the anterior shell margin at the pedal gape; posteriorly it forms the wide, shallow pallial sinus.

## Mode of life

*Laternula elliptica* lives completely buried in a vertical position within muddy and sand-muddy substrata of the sea bottom (Fig. 2); underwater *in situ* photos showed that only few centimeters of the siphonal distal end in larger specimens are extended into the water column. All living specimens observed in aquaria ($n = 20$) were able to rebury, the smallest ones performing such activities much faster. Thus, while nine individuals whose shell length ranged from 1.0 to 3.4 cm were found totally buried after six hours of being placed in aquaria with muddy sediment, the eleven larger specimens (shell length 4.0 to 9.6 cm) took up to three weeks to accomplish the same task. Only a few individuals in the latter group exhibited "jetting movements" (*sensu Ansell & Rhodes, 1997*) on the sediment surface (Figs. 2A and 2B). In contrast to the reported observations of *Ansell & Rhodes (1997)* and *Peck et al. (2004)*, these specimens did not try to burrow at the end of each cycle of movement. Likewise, additional "looping" and "levering" movements as described by these authors were not observed during the short research period of this project.

The siphons play an important role in the burrowing process. Individuals with their shells completely buried and with the reduced foot anchored in the substratum, force the wall of the siphons and the shell valves tightly against the sediment (Fig. 2C). This is accomplished by raising the hydrostatic pressure within both the pallial chamber and siphons through the closure of the pedal and siphonal openings, followed by a slow retraction of the siphons and concomitant relaxation of the orbital (pallial) and adductor muscles. Further vigorous retraction of the still-closed siphons, followed by contraction of the adductors and orbital muscles, and the opening of the pedal aperture force water to be powerfully expelled through the pedal aperture only. Jetting removes sediment from the depths of the burrow as the water exits through a narrow gap between the animal and the surrounding sediment (two asterisks in Fig. 2C). Subsequent contraction of the pedal retractor muscles pull the cylindrical animal deeper into the hollow excavated below the animal. When disturbed (by using forceps), some of the largest (5 to 9.6 cm in shell length) and two small (±2.0 cm in shell length) buried individuals kept the siphonal walls so tightly pressed against the surrounding sediment that the water jet drilled a tunnel through the substratum and ejected mud particles into the water column as it exited the substratum a short distance away from the bivalve (one asterisk in Fig. 2C). The effect of such a muddy "spring" on the sediment surface can be seen in Fig. 2D, a photograph taken while SCUBA diving in the natural habitat.

## Mantle

The mantle lobes are thin and translucent, except at their muscular border where the strong pallial muscles are inserted to and unite both valves.

The mantle margins are extensively fused, except for the small, anteroventral pedal gape and the posterior inhalant and exhalant siphonal openings (Figs. 1B, 1E, 1F and 3). There

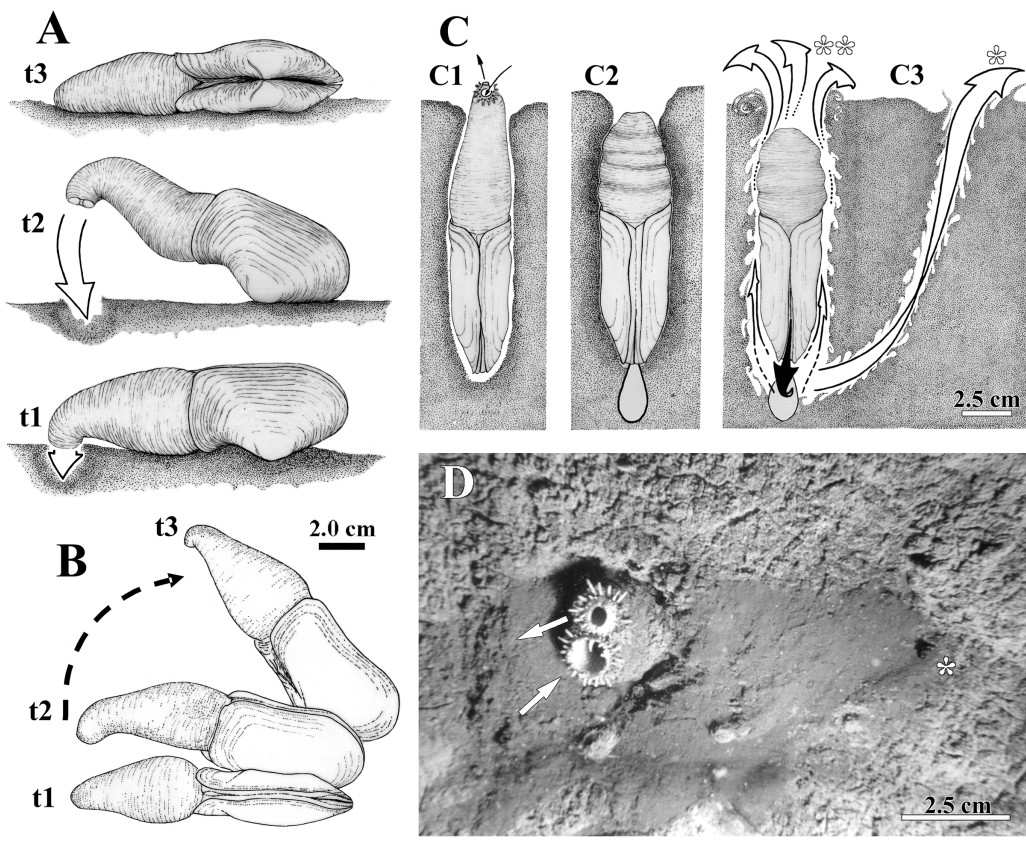

**Figure 2 Observed behavior of *Laternula elliptica*.** (A and B) Surface movement ("jetting" cycle): (A) viewed from sidewall of the aquarium. (B) Same, viewed from water surface. In "t1" the animal is lying on the sediment surface, dorsal side down. The initial phase of the cycle is preceded by the closure of the pedal gape, valves opening, and swelling of both siphons that bend their tips onto the sediment surface. In "t2" the adductors and orbital muscles contract and the diameter of the siphons reduces, generating a strong jetting (arrow); only the posterior half of the shell and siphons are lifted above the sediment surface, while the body rotates around its antero-posterior axis. In "t3" the cycle completes with the animal lying on one shell valve, after a clockwise/anticlockwise translocation (arrow in B) of the animal. (C) Burrowing behavior: (C1) Ventral view of the animal in its natural position, with the arrows indicating inhalant and exhalant currents. (C2) Protective reaction against predators, with the animal closing pedal and siphonal openings, relaxing pallial and adductor muscles, and retracting siphons; positive hydrostatic pressure generated by the water in the pallial chamber and siphons forces the valves and siphonal walls tightly against the sediment, preventing collapse of the surrounding soft, plastic sediment. (C3) Burrowing within the substratum: to move deeper into the substratum, the animal contracts the siphons and expels water vigorously through the pedal opening (black arrow), stirring and removing sediment from the depths of the gallery. White arrows indicating the two escape routes for the water: running through the narrow space between the shell and sediment (indicated by two asterisks in C); and drilling a tunnel throughout the sediment to emerge a short distance from the bivalve (indicated by one asterisk in C3 and D). Under gravitational forces or by contraction of the pedal retractor muscles, the heavy and cylindrical body "drops" into the hollow excavated below the animal. (D) Underwater photograph taken just after complete precipitation of the blackish mud removed from the substratum during burrowing activity. The distal end of the siphons is exposed above the sediment surface; arrows indicating inhalant and exhalant currents.

is no 4[th] pallial aperture. From the mantle isthmus, fusion extends forward up to the dorsal edge of the anterior adductor muscle, and posteriorward up to the base of the exhalant siphon; it involves both the inner and middle mantle folds, as well as the periostracal grooves (type C of *Yonge, 1957*). Fusion in these regions accounts for the formation of an extensive secondary ligament that unites the shell valves dorsally (Figs. 1D, 1G and 1H). From the dorsal edge of the anterior adductor muscle downward to the dorsal edge of the pedal opening, mantle fusion involves the inner folds and the inner surfaces of the middle folds only (type B of *Yonge, 1957*). This same type of fusion occurs along the entire extent of the ventral margin between the pedal opening and the base of the inhalant siphon, and accounts for the presence of a sheet of periostracum lining each side, except along the median longitudinal line of fusion. The pallial muscles along this ventral margin extend from one to the opposite valve and form the orbital muscles as termed by *Morton (1976)* in *Exolaternula spengleri* (Gmelin, 1792) (as *Laternula truncata*). The orbital muscles in *L. elliptica* act as a long, accessory ventral adductor as it was demonstrated experimentally: after having the orbital muscles separated from one or both valves, living specimens ($n = 2$) with the adductor muscles and shell valves intact were unable to bring the ventral border of the valves in close contact. Likewise, specimens collected along ice scours within the Admiralty Bay and with one or both of their valves severely damaged ($n = 4$) could tightly close the pieces of the shell adhering to the orbital muscles, even though these fragments were not under the control of the adductors.

## Siphons

The conjoined siphons of *L. elliptica* are formed by fusion of all three marginal mantle folds including the periostracal groove (type C of *Yonge, 1948*, *1957*, *1982*), which accounts for the thick, corrugated, brownish periostracum that covers the siphonal walls (Fig. 3B). Fully extended siphons reach almost twice the shell length, as observed in a non-buried, 9 cm shell length specimen that extended its siphons up to 14 cm; although their diameter equals that of the animal's body, they are capable of a slow, but complete retraction into the shell.

During siphoning, the tips of the siphons are the only parts kept in the water column. Not infrequently, freshly collected specimens had these parts of the siphons fouled (and thus camouflaged) by living hydrozoans, bryozoans, and filamentous algae attached to the periostracum. Such epibionts and other extraneous elements from the surrounding sediment are firmly adhered to the surface of the periostracum by fine threads of a sticky secretion exuded from the apex of rounded papillae. These papillae form a continuous line adjacent to and internal to the periostracal groove surrounding the siphonal apertures (Fig. 4A). Each papilla corresponds to the discharging point of an arenophilic mantle gland, as shown by *Sartori, Passos & Domaneschi (2006)*, who studied these glands in specimens of *L. elliptica* collected in the same field study.

The distal tips of both inhalant and exhalant siphons bear a crown of numerous digitiform tentacles (Fig. 4A); four to nine tentacles on the inhalant, and five to seven on the exhalant siphon, bear a complex eye at their distal end (optic tentacles) (Fig. 4B). The eyes (Fig. 4C) have structure and complexity similar to those described by *Morton*

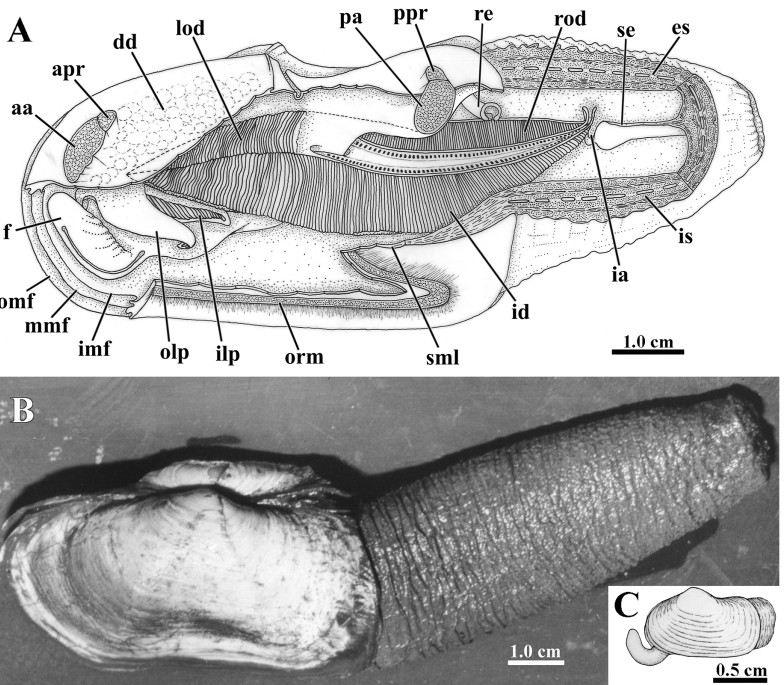

**Figure 3** *Laternula elliptica*—**anatomy, with focus on the pallial cavity (A), and external morphology (B and C).** (A) Organs of the pallial cavity viewed from the left side, after removal of the left shell valve and partial section of the left mantle lobe, outer demibranch and siphons (Details of the ctenidial and labial palp folds and ciliary currents are shown in Figs. 6 and 7). (B) Photograph of a living specimen from the left side. (C) Sketch of a juvenile. Abbreviations: aa, anterior adductor muscle; apr, anterior pedal retractor muscle; dd, digestive diverticulum; es, exhalant siphon; f, distal portion of the foot; ia, inter-chamber aperture; id, inner demibranch; ilp, inner labial palp; imf, inner marginal mantle fold; is, inhalant siphon; lod, left outer demibranch; mmf, middle marginal mantle fold; olp, outer labial palp; omf, outer marginal mantle fold; orm, orbital muscle; pa, posterior adductor muscle; ppr, posterior pedal retractor muscle; re, rectum; rod, right outer demibranch; se, inter-siphonal septum; sml, sectioned mantle lobe.

*(1973)* and *Adal & Morton (1973)* for *Exolaternula spengleri* (as *L. truncata*). Neither regular number nor arrangement of the tentacles could be identified, but as a rule, they enlarge in size centrifugally, the optic tentacles being amongst the largest ones. Scarce tactile tentacles occupying an outer position in the crown, with each bearing a distal black pigment spot that looks like an ill-defined eye.

In addition to the crown of tentacles at its periphery, the inhalant aperture has its free border indented by a series of digitiform tentacles of three different orders of size (Fig. 4A). As a general rule, four to six longer, first order tentacles alternate regularly with four to six medium-sized, second order tentacles. Inserted in between the first and second order tentacles lie one to three short, third order tentacles. Some first order tentacles are bifid.

The inhalant aperture contracts and expands quite uniformly, thus suggesting it is provided with a circular sphincter of muscular fibers. The tentacles associated with this aperture can be brought either closer or further, as well as bent either centrifugally, allowing free intake of water and suspended material, or centripetally, creating a barely functional barrier against large particles and excess of material.

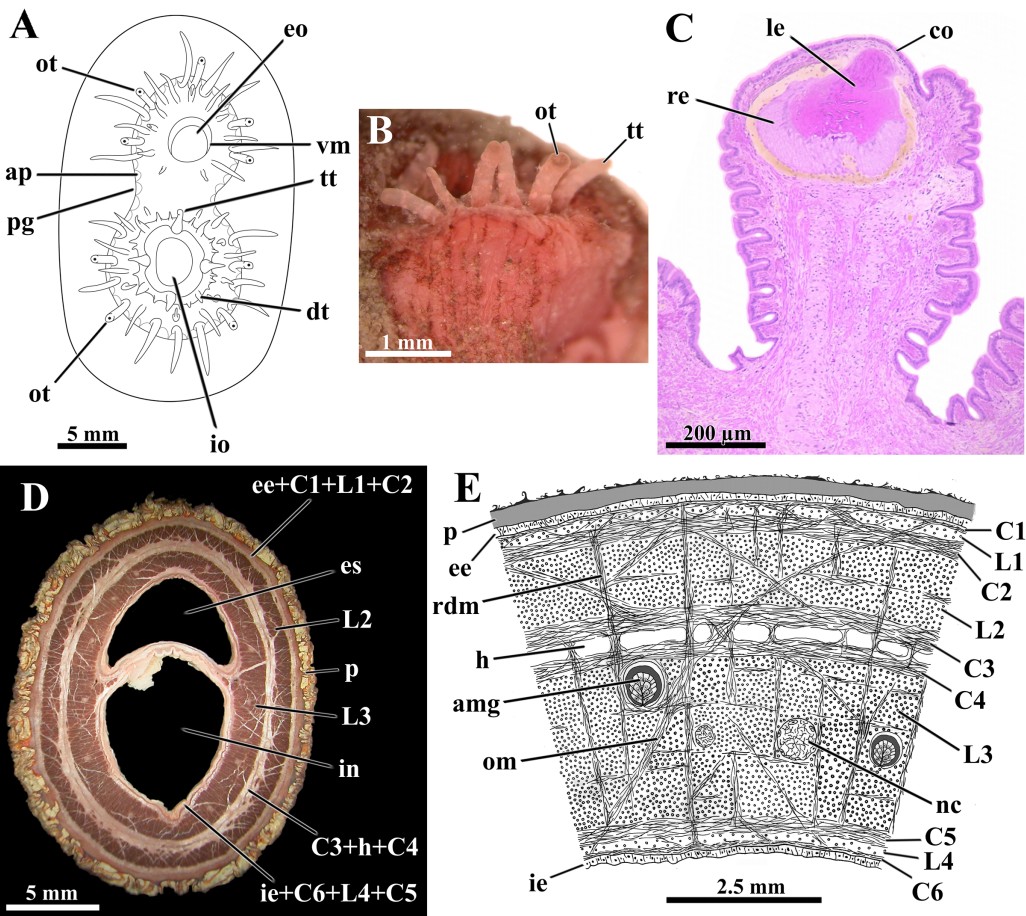

**Figure 4** *Laternula elliptica*—**siphons.** (A) Frontal view of the siphonal tips (the lines of arenophilic threads with adhered particles in the siphonal walls here omitted for simplification). (B) Fragment of the siphonal end, showing optic and tactile tentacles. (C) Sagittal section of an optic tentacle. (D) Transverse section of the siphonal walls, showing the musculature, and thick periostracal covering. (E) Diagrammatic transverse section through the wall of the conjoined siphons near their tips. Abbreviations: amg, arenophilic mantle gland; ap, arenophilic papilla; C1 to C6, circular muscle layers; co, cornea; dt, digitiform tentacle of the inhalant aperture; ee, external epithelium; eo, exhalant opening; es, lumen of exhalant siphon; h, haemocoel; ie, internal epithelium; in, lumen of inhalant siphon; io, inhalant opening; L1 to L4, longitudinal muscle layers; le, lens; nc, nerve cord; om, oblique muscle strands; ot, optic tentacle; p, periostracum; pg, periostracal groove; rdm, radial muscle strands; re, retina; tt, tactile tentacle; vm, valvular membrane.

The exhalant aperture lies at the summit of a thin, smooth, volcano-shaped valvular membrane (Fig. 4A). Similar to what was described by *Morton (1973)* in *Exolaternula spengleri* (as *L. truncata*), this aperture closes by contraction at two opposite lines of folding, one dorsal and one ventral, thus forming two lateral valves. The fully expanded valvular membrane is maneuvered around the siphon axis, driving the exhalant current with rejected material and gametes far from the inhalant aperture.

Irregular bands of brown and yellowish-white pigment delicately pattern all tentacles and the epithelium circumscribed by the periostracal groove. A homogeneously dispersed light-green pigmentation, as well as patches of brown pigment that fade away onto the base of the siphons, are also present all over the inner epithelium of both organs.

The wall of both siphons is provided with a thick musculature (Fig. 4D). This is arranged, from the outer to the inner epithelium, in the following muscle layers (Fig. 4E ): a narrow circular layer (C1), intermingled with isolated bundles of longitudinal fibers (L1); a thick circular layer (C2); a thick longitudinal layer (L2); two central circular layers (C3 and C4) separated by a haemocoel; a massive longitudinal layer (L3) containing the nerve cords; a thick circular layer (C5); a narrow band of isolated bundles of longitudinal fibers (L4); and a circular layer (C6) adjacent to the inner epithelium. Radially arranged muscle strands run from one epithelium to the other, splitting the longitudinal muscle layers L2 and L3 into a series of sharply defined bundles, and the haemocoel lying between C3 and C4 into a linear series of compartments. Ubiquitous oblique muscle strands arising from the circular muscle layers similarly cross the muscular layers. Adjacent to each opposite margin of the intersiphonal septum lies a wide, longitudinal haemocoelic compartment.

At the base of the siphons and inserted in the longitudinal layer L3 there are fourteen nerve cords, six in the exhalant and eight in the inhalant; these cords ramify as they extend toward the tip of the siphons, where up to 24 nerves were identified.

The septum that divides the inhalant from the exhalant lumina is membranous, poor in muscular fibers and extremely flexible at its basal portion near the posterior end of the ctenidia. It thickens toward the distal end of the siphons, as the muscular layers C6, L4, C5, L3, and oblique muscle strands participate in its constitution. Retraction of the siphons is accomplished by vigorous contraction of the longitudinal muscles whereas protraction requires the modulation of the radial and circular muscles acting on the haemal fluid.

## Musculature and foot

The epithelium that lines both the distal and proximal (= visceral) portions of the foot bears 5 µm-long cilia; however, ciliary currents were detected on the visceral portion only. The distal, muscular portion of the foot is roughly hatchet-shaped and small (±1/6 of the shell length) when contracted; fully extended it reaches ±1/4 of the shell length. When protracted, the distal portion can extend to a reasonable distance beyond the shell margin and function as a digging tool, even in the largest specimens; juveniles possess a comparatively longer and more mobile foot (±1/2 the shell length in 2.0-cm-long specimens) (Fig. 3C).

A shallow, vestigial byssal groove is easily seen along the ventral edge of the contracted foot, but quite indiscernible when it is elongated. At its rear end there opens a single ciliated duct that bifurcates to join with the right and left components of a vestigial byssus gland embedded in the visceral portion of the foot.

The general muscular system of L. elliptica is shown in Fig. 5. The anterior and posterior adductor muscles are reduced, with elliptical, subequal insertion areas. The extrinsic pedal musculature consists of bilateral pairs of small anterior and posterior pedal retractors, and one pair of anterior pedal protractors. Though both pairs of retractors have similar insertion area, the anterior pedal retractors are thicker than the posterior ones.

The anterior pedal retractors attach to the shell valves close to and behind the dorsum of the anterior adductor muscle; thence, both the right and left muscles pass downward almost vertically, flatten and twist as they converge to and unite at the sagittal plane just

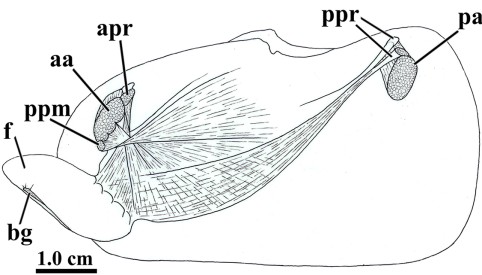

**Figure 5** *Laternula elliptica—m*usculature, as viewed on the left side. Abbreviations: aa, anterior adductor muscle; apr, anterior pedal retractor muscle; bg, byssal groove; f, distal portion of the foot; pa, posterior adductor muscle; ppm, pedal protractor muscle; ppr, posterior pedal retractor muscle.

below the esophagus. At this point, their fibers spread out and penetrate both the proximal (visceral) and distal portions of the foot, where they form the innermost muscular layer of the organ.

The posterior pedal retractor muscles flatten and thin as they extend anteroventrally and unite under the kidneys; from here, their fiber bundles become well discernable as they spread fanwise at the ventrolateral sides of the visceral mass and form a muscular layer external to that of the anterior pedal retractors.

The pedal protractor muscles are the most developed among the extrinsic muscles. The main fiber bundle inserts on the shell valves juxtaposed ventrally to the anterior adductor muscle; thence, this bundle extends horizontally and posteriorward as it twists and spreads out on the dorsal half of the proximal (visceral) portion of the foot. The remaining, less developed portion of the protractor penetrates shallowly into the posterior side of the anterior adductor muscle and inserts on the shell valves with the adductor; its fibers forming a thin layer as they spread out ventral- and posteriorward on the ventral half of the proximal (visceral) portion of the foot.

In addition to the extrinsic pedal muscles, the visceral and distal portions of the foot are supplied with isolated, transverse muscle strands (intrinsic pedal musculature), which insert on the cubical epithelium lining each side of the foot.

## Ctenidia

The long, deeply plicate, eulamellibranch and heterorhabdic ctenidia of *L. elliptica* extend from the labial palps deep into the siphons, well beyond the posterior limit of the shell in specimens with protruded siphons (Fig. 3A). Each inner demibranch comprises descending and ascending lamellae of near-equal height and bears a deep marginal food groove; the outer demibranch consists solely of an upturned descending lamella (Fig. 6A).

The number of filaments per plica varies along the ctenidia of all specimens and increases with age. The ordinary filaments form the bulk of each plica, bearing frontal, latero-frontal and lateral cilia (Figs. 6B–6D). Three (occasionally two) filaments at the apex of each plica (Figs. 6B, 6D, 6E and 6G) are higher, with a broader frontal surface and a larger number of mucocytes than the ordinary filaments on the sides.

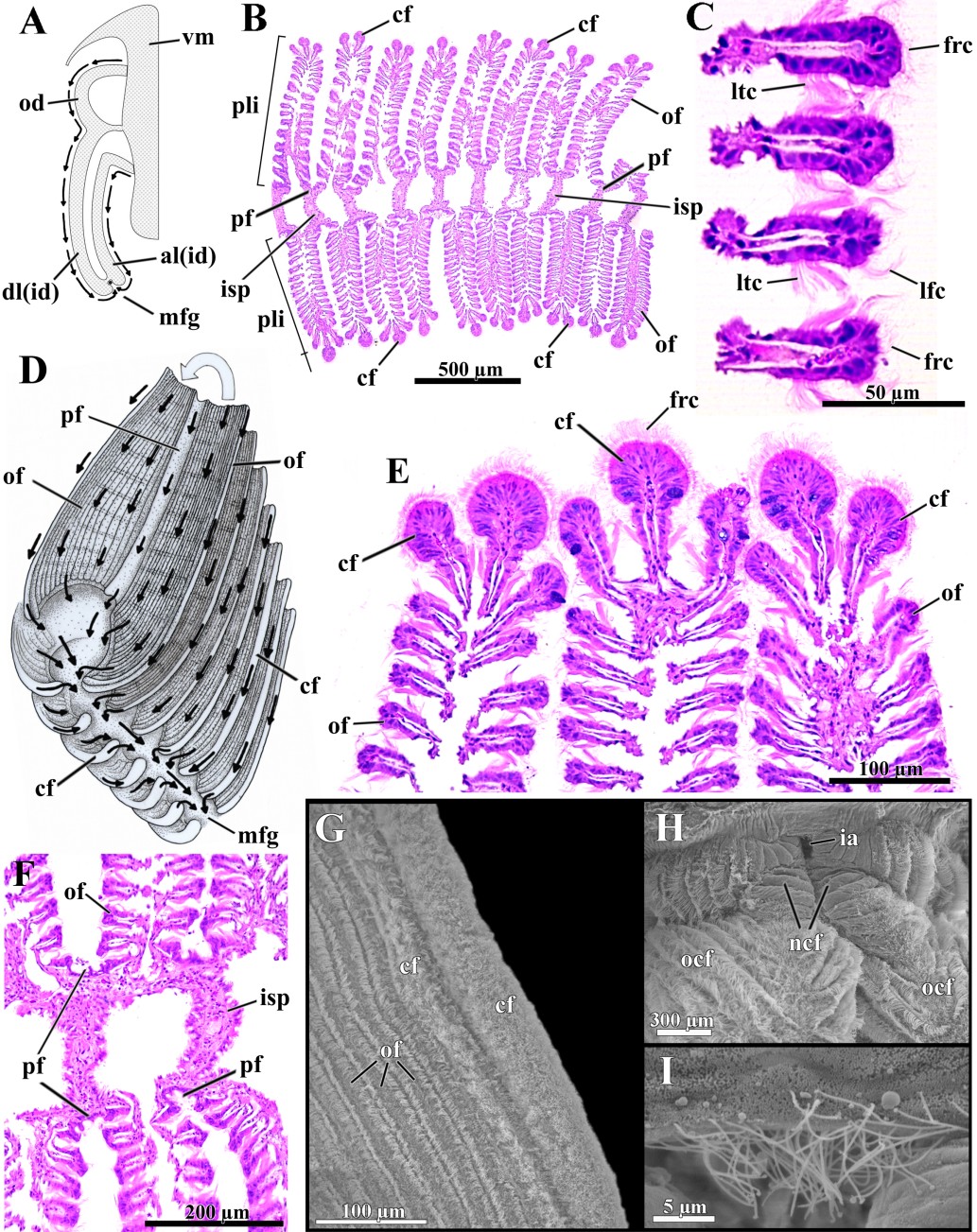

**Figure 6** *Laternula elliptica*—**ctenidia.** (A) Transverse section, diagrammatic view of the ctenidial ciliary currents. (B) Histological section of the inner demibranch, showing the plica with their ordinary, crest, and principal filaments. (C) Detailed view of four ordinary filaments. (D) Detailed sketch of the ctenidial filaments, with one fold turned out to expose principal and ordinary filaments. (E) Detailed view of the top of three plicae, showing the crest filaments. (F) Detailed view of four principal filaments with their respective intraplical septa. (G) Scanning electron micrograph of a plica. (H) Scanning electron micrograph of inter-chamber aperture of a juvenile (1.0 mm long) specimen. (I) Detailed view of the cilia bordering the aperture in (H). Abbreviations: al(id) and dl(id), respectively, ascending and descending lamella of the inner demibranch; cf, apical (crest) filament of plica; frc, frontal cilia; ia, inter-chamber aperture; isp, intraplica septum; lfc, latero-frontal cilia; ltc, lateral cilia; mfg, marginal food groove; ncf, newly formed ctenidial filaments; ocf, older ctenidial filaments; od, outer demibranch; of, ordinary filament; pf, principal filament; pli, plica; vm, visceral mass.

At regular intervals, interfilamentar junctions expand across the intraplical space and form complete intraplical septa; these septa lie parallel to each other and compartmentalize the full extent of the intraplical space in both demibranchs.

The principal filaments are clearly differentiated, with a broad, shallow U-shaped frontal surface (Figs. 6B and 6F). The abfrontal surface of every other pair of principal filaments in the inner demibranchs fuses into a complete, high interlamellar septum that almost reaches the ctenidial axis; these high septa alternate with low interlamellar septa that extend but a short distance from the free, ventral margin of the inner demibranchs.

The abfrontal portion of all principal filaments of the outer demibranchs forms a low-extended septum that does not attach to the epithelium of the visceral mass. Thus, at each side of the body the outer demibranch and the epithelium of the visceral mass define a narrow compartment that is continuous with the spacious suprabranchial chamber lying posterior to the visceral mass.

The free ventral tips of the plicae that form the inner demibranchs give a deeply scalloped appearance to the walls of the marginal food groove (Figs. 6A and 6D), which can move toward and away from one another, acting as a sorting device.

The frontal ciliary currents on both demibranchs are exclusively toward the ventral, marginal food groove (Figs. 6A and 6D) and the ctenidia can thus be ascribed to *Atkins (1937)* type E. Sorting mechanisms all over the outer and inner demibranchs are of the "*Pinna* type" of *Atkins (1937)*, *i.e.*, fine particles traveling along the grooved frontal surface of the principal filaments and on the frontal surface of their adjacent ordinary filaments are passed to an active oralward current within the ventral marginal food groove, whereas coarse and excess particles traveling on the remaining lateral and apical filaments are transferred to an oralward current outside the marginal food groove and rejected.
The ctenidia are highly muscular and very sensitive; if stimulated, the plicae both shorten and flatten locally. By adjusting the distance both among plicae and lateral walls of the marginal food groove, the animal can further regulate the oralward uptake of particles. The plicae and lateral walls of the food groove hide the main acceptance tracts and expose unwanted and excess particles to an entirely rejectory surface. Fine particles only and thin mucous strands protected inside the marginal food groove are carried toward the mouth; this is the only oralward current along the ctenidia.

The dorsal margin of the ascending lamella of each inner demibranch forms a translucent membrane that attaches to the visceral mass by cuticular fusion; posterior to the visceral mass the ctenidial axes hang free and the membranous margins of both ascending lamellae unite each other by tissue fusion, forming the floor of the spacious, posterior portion of the suprabranchial chamber. The dorsal margin of the upturned outer demibranchs is also attached to the visceral epithelium by cuticular fusion. Cuticular fusion in *L. elliptica* is not easily detached in living or preserved specimens; it resists both displacement of the inner and outer demibranchs and strains at the inner membranous margins of the inner demibranchs.

The posterior end of both ctenidial axes and inner demibranchs do not fuse with the inter-siphonal septum, leaving a direct, permanent communication between the supra- and infrabranchial chambers (Figs. 3 and 6H) that was termed "inter-chamber aperture"

by *Sartori & Domaneschi (2005)* in *Thracia meridionalis*. The free tips of the ctenidial axes form two tentacular projections that bend either dorsalward into the suprabranchial chamber or retract ventrally through the inter-chamber aperture. The membranous, basal portion of the inter-siphonal septum expands into a flat, trigonal lip that acts as an efficient valve allowing the animal to either retract and tightly close the inter-chambers aperture or expand it widely. The aperture widens as the inter-chamber valve swells out ventrally into an igloo-shaped structure, with its free ciliated border (7.5 μm-long cilia; Fig. 6I) taking a U-shape outline. Conversely, flattening the domed valve up, its free, ciliated border is pushed forward and inserted in between the rear end of the ctenidia, thus isolating the infra- from the suprabranchial chamber completely. In its flattened state, the valve and inter-chambers aperture are easily overlooked; however, both are present from early juvenile stage as it could be confirmed by SEM of a minute, 1.0 mm-in-shell-length specimen (Figs. 6H and 6I), as well as by careful dissections of living and well-preserved specimens measuring 1.0 through 9.6 cm in shell length. The ability to route water from the supra- to the infrabranchial chamber was tested in seven living specimens (1.0 through 8.0 cm in shell length). The animals had their exhalant siphon lumen injected with a concentrated carmine suspension and immediately stimulated with forceps both to contract and tightly close the exhalant opening. Water jets containing carmine particles were observed leaving forcibly through the pedal opening of five specimens and through both the pedal and inhalant openings of two, thus corroborating data from the morphology.

## Labial palps and lips

The labial palps are long (one fourth of the shell length), triangular, with the folded surfaces framed by a wide, smooth area on both dorsal and adoral sides, and a narrow one along the ventral side of the organs (Fig. 7A). Very sensitive to mechanical stimuli, the palps may either roll up longitudinally into a hollow cone with the ventral and dorsal margins touching each other, or coil up spirally; in both cases the folded surface faces outward (Fig. 7A). The palps can also expand/contract moving their numerous low folds apart or closer; the folds can also either bend oralward or stand quite upright, thus hiding or exposing the troughs between them.

Figure 7 shows the structure and ciliary sorting mechanisms on the palp surfaces (currents "a" through "i"). Transversely dorsalward current (a), on the smooth outer surface, conveys particles onto the smooth dorsal area of the folded surface. Thence, particles may be either thrown downward (b) toward the plicae or be captured and transported to the subdistal free end of the palp by a longitudinal ciliary tract (c); cilia on this portion transfer material to the folded area. Transversely directed currents (d) operating oralward across the crests of the folds act as acceptance or rejection currents, depending on the size and/or total volume of particles. Cilia on the crests transfer (i) excess material and/or large particles onto a powerful rejection ciliary tract (e) along the narrow, smooth ventral margin of the organ; fine material trapped on the dorsal half of the plicae is preferably transferred to the mouth. Ciliary tracts (f) on the oral surface of each plica deliver isolated particles either onto a rejection tract (g) on the floor of the groove between

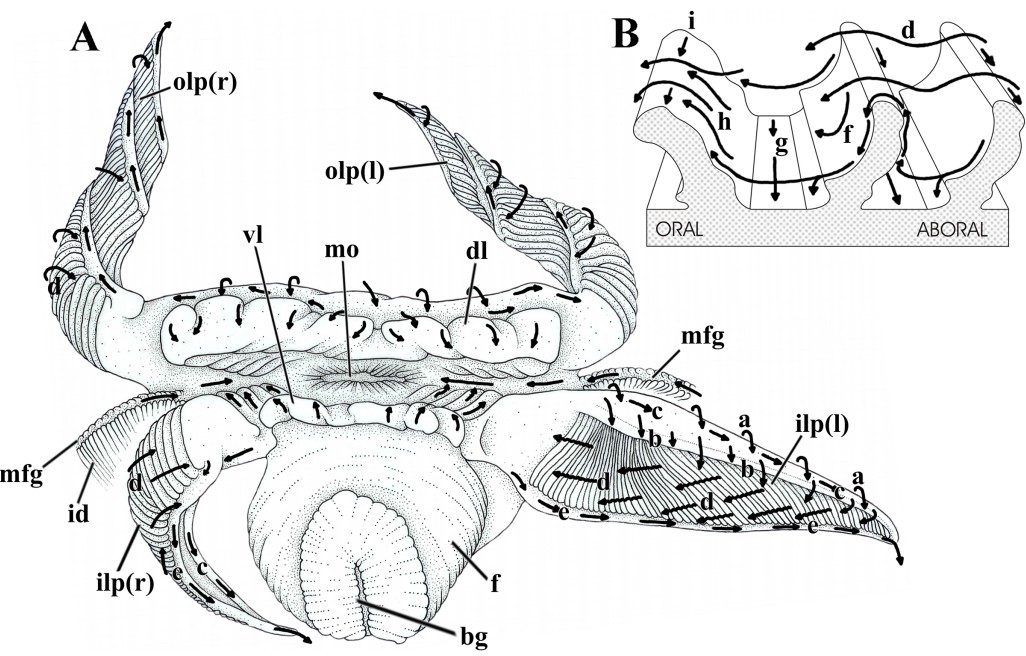

**Figure 7 _Laternula elliptica_—labial palps.** (A) Semi-diagrammatic anterior view of the oral region, summarizing the functioning of the labial palps. Outer palps shown coiled spirally; the right inner one bent longitudinally, the left inner one extended. (B) Diagrammatic section through three folds of a palp, showing ciliary currents. Abbreviations: a through i, ciliary currents (see text for details); bg, byssal groove; dl, dorsal lip; f, foot; id, inner demibranch; ilp(r) and ilp(l), inner right and left labial palp, respectively; mo, mouth; mfg, marginal food groove; olp(r) and olp(l), outer right and left labial palp, respectively; vl, ventral lip.

adjacent folds, or onto the aboral surface of its anterior, adjacent fold; here, ciliary tracts (h) transfer both large and minute mineral and organic material onto currents "d". Along the ventral third of the palps, particles traveling on currents "h" are intercepted by longitudinal ciliary tracts (i) on the aboral side of the crests and transferred to the main rejection tract "e" along the free ventral margin of the palp. Particles present on currents "g" also converge to this rejection tract "e".

In addition to regulating the intake of particles by adjusting the steepness of the folds and/or the distance between them, _L. elliptica_ can further regulate the amount of material being carried oralward by strengthening the rejection currents in two ways. The labial palps roll up longitudinally, bringing together both their dorsal and ventral margins and their respective longitudinal currents "c" and "e", which convert into a strong rejection current that sweeps away unwanted and excess material coming into contact with the folded surface (Fig. 7A, right inner palp). Alternatively, spiral coiling of the palp (Fig. 7A, both right and left outer palps) brings the rejection ciliary tract "e" into intimate contact with the folded surface; being stronger, the rejection current "e" intercepts and gets rid of excess material being directed oralward on currents "d".

The long and wide dorsal and ventral lips deal with isolated particles that go deep into the anterior region of the mantle chamber. Both have the inner surface with a flat, distal margin, more conspicuous in the dorsal lip, and a cushion-like, often transversely

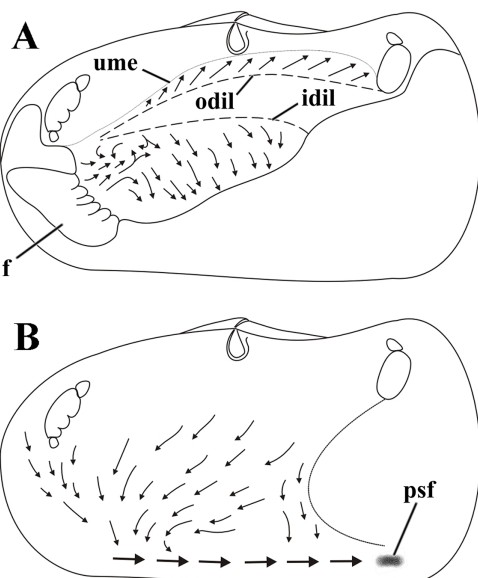

**Figure 8** *Laternula elliptica*—**cleansing ciliary currents, indicated by arrows.** (A) Currents on visceral mass surface. (B) Currents on inner surface of right mantle lobe. Abbreviations: f, foot; idil and odil, lines of insertion of the inner and outer demibranchs with the visceral mass, respectively; psf, pseudofaeces; ume, upper mantle edge (roof of the pallial cavity).

corrugated basal portion. Corrugations may either mimic transverse folds or disappear as the lips contract and relax, respectively. Transversely directed cleansing currents on the flat, smooth outer surface of both lips convey particles onto their inner surfaces; thence, particles are passed transversely onto the oral groove; on the dorsal palp they may also be trapped by a ciliary tract that delivers unwanted material to the rejection current "e" along the free ventral margin of the palps.

## Ciliary currents on the visceral mass and inner mantle surface

Weak ciliary cleansing currents on the visceral mass epithelium sweep particles ventral- and posteriorward (Fig. 8A), except at its anterior portion overlapped by the proximal third of the inner labial palps; in this anterior portion particles are carried dorsalward and caught by cilia on the smooth outer surface of the palps and passed to the folded surface of this organ to be resorted. Unwanted material about to reach the ventral limit of the visceral mass either falls onto the rejection currents of the mantle or is removed by frontal cilia of the ctenidia and ultimately discarded to and rejected by the mantle.

Cilia on the visceral mass epithelium, dorsal to the line of attachment of the reflected outer demibranch, sweep particles dorsalward, toward the mantle lobe surface.

Ciliary activity all over the inner mantle surface transfers particles ventral- and anteriorward onto the posterior end of the pedal opening predominantly (Fig. 8B). Here, a single, strong rejection tract receives the bulk of pseudofeces coming also from the ctenidia, labial palps and visceral mass epithelium and drives it posteriorward and concentrates in large mucous masses at the base of the inhalant siphon. Unwanted material so collected is periodically ejected through the inhalant siphon.

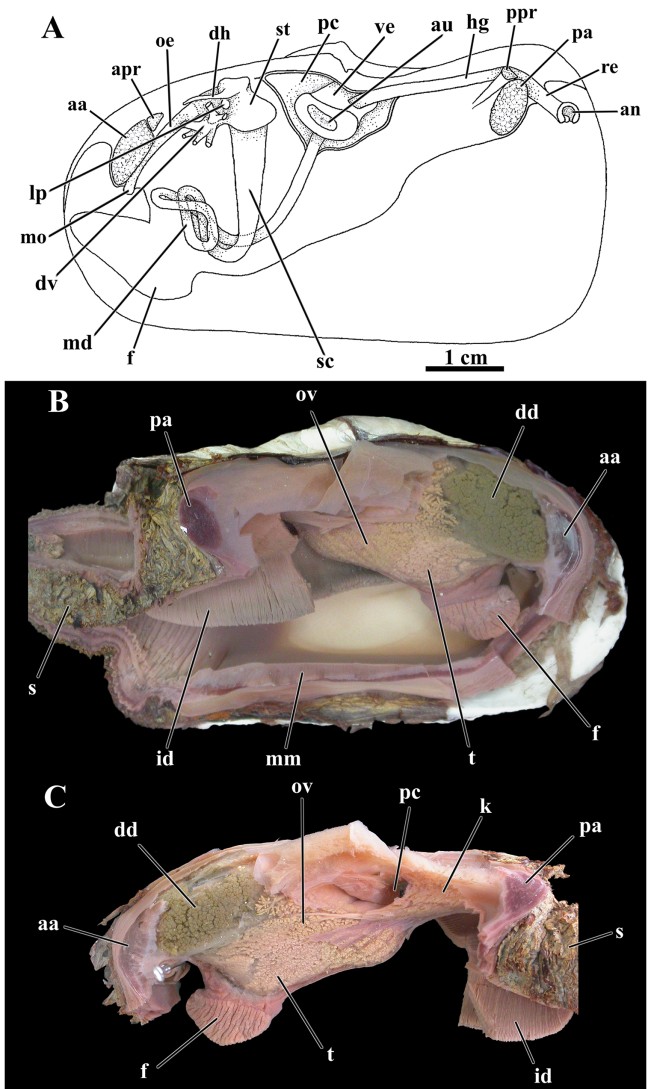

**Figure 9** *Laternula elliptica*—**organs of the visceral mass.** (A) As viewed from the left side. (B) Right side, after excision of most of the mantle, siphons, gill, and labial palps. (C) Left side, after removal of these organs (except for a small part of the siphons and the inner demibranch). Abbreviations: aa, anterior adductor muscle; an, anus; apr, anterior pedal retractor muscle; au, auricula; dd, digestive diverticulum; dh, dorsal hood; dv, ducts of the digestive diverticulum; f, foot; hg, hindgut; id, inner demibranch; k, kidney; lp, left pouch; md, midgut; mm, mantle margin; mo, mouth; oe, oesophagus; ov, ovarium; pa, posterior adductor muscle; pc, pericardial cavity; ppr, posterior pedal retractor muscle; re, rectum; s, siphons; sc, style sac; st, stomach; t, testes; ve, ventriculum.

## Digestive tract

The mouth (mo) and lips are positioned adjacent and posterior to the ventral border of the anterior adductor muscle and the long, flattened tubular oesophagus opens into a conical, anterior projection ("vestibule") of the globular portion of the stomach (Fig. 9A). A conical-shaped dorsal hood, as long as the oesophagus (1/6 of the shell length), projects from the roof of the stomach and curves over toward the left side (Fig. 9A). This pocket has a short, swollen proximal portion and a five-times-longer slender portion extending

parallel and adjacent to both the vestibule and the posterior half of the oesophagus. Ventral to the swollen portion of the dorsal hood arises a small, shallow blind pouch (left pouch), and from both sides of the floor of the stomach emerge the ducts that connect with the right and left digestive glands (Figs. 9A and 9B). The combined style sac and mid gut extend almost vertically downward from the posterior, ventral wall of the globular portion of the stomach. The distal third of the style sac bends smoothly anteriorward and finishes without any striking modification in its diameter as it turns into the isolated mid gut. The midgut bends anteriorly and then dorsalward as it leaves the style sac and performs a series of three to five short, tight loops just anterior to the distal half of the style sac. Thence, the midgut bends ventralward, passes by the right side of the style sac and curves dorsalward to pass through the pericardial cavity, ventricle and among the kidneys ducts before finishing in the anus. The rear end of the intestine extends further posteriorly to the adductor muscle as a free, maneuverable extension of the rectum. Examination of the gut content found amorphous organic matter particles, which (according to O. Domaneschi's, 1996–1998, observations) originated from the water column or from the layer just above the surface sediment.

### The pericardial cavity and kidney

An ample pericardial cavity lies under the umbos and contains the heart formed of a large ventricle and two flat, lateral auricles, with the ventricle penetrated by the hind portion of the intestine (Fig. 9C). Two renopericardial apertures located at the floor of the pericardial cavity drain primary urine onto the proximal arms of the kidney. The bulk of the kidney is wedged between the pericardial cavity and the anterior face of the posterior adductor muscle with the chief axis almost horizontal; the hind gut lays dorsally on the mass of renal tissue. Each distal arm of the kidney fuses with the distal end of the hermaphrodite duct (see below) before discharging into the suprabranchial chamber *via* a common urino-genital opening.

### Gonads

As was found for other previously studied anomalodesmatans, *L. elliptica* is a hermaphrodite, with a pair of testes occupying a ventral position in the visceral mass, more concentrated around and among the intestinal loops, and a pair of well-developed ovaries packed mainly on the posterior half of the visceral mass, thence expanding forward both around and amongst the digestive diverticula (Figs. 9B, 9C).

From each testis arises a long vas deferens that passes postero- and dorsalward onto the floor of the kidney. Here, the vas deferens presents a minute opening communicating with a very short, thin-walled fragile oviduct and extends farther posterior as a hermaphrodite duct. Close to its rear tip the combined, male-female duct receives the opening of the distal arm of the kidney and both reproductive and excretory systems discharge at the urino-genital papilla within the suprabranchial chamber.

During the austral summers of 1996/1997 and 1997/1998, specimens with shells larger than 5.0 cm had both the ovaries and testes crowded with ripe eggs and mobile sperm cells, respectively. As the animals were manipulated for morphological studies, live, dissected

specimens laid thousands of eggs adhered to each other in linear, short threads that precipitated quite immediately onto the bottom of the Petri dishes.

## DISCUSSION

The Anomalodesmata comprises a diverse group of bivalves, with the members of the Laternulidae being well known as having a sedentary mode of life, living deeply burrowed intertidally or sublittorally. Although comprising a relatively small number of species, the taxonomy of the living species of Laternulidae has been much confused and discussed in the literature (*e.g.*, *Huber, 2010*, *2015*; *Prezant, Shell & Wu, 2015*). A preliminary revision by *Taylor et al. (2018)*, based on molecular data, museum specimens, and literature data, grouped the approximately 15 extant taxa of the family into two genera, *Laternula* Röding, 1798 and *Exolaternula Habe, 1977*, and pointed to several synonymies and misidentifications in prior publications that have covered the members of the group. This is of relevance in the current context as the few existing morpho-anatomical data in the literature were assigned, in part, to incorrect nominal taxa.

*Exolaternula* differs from *Laternula* in having a lithodesma present in the adult, with *Taylor et al. (2018)* recognizing three valid species in this genus, *E. spengleri* (Gmelin, 1792), *E. liautaudi* (Mittre, 1844), and *E. erythraea* (Morris & Morris, 1993), and about a dozen species in *Laternula*. *Habe (1977)* stated the type species of *Exolaternula* to be *Anatina truncata* Lamarck, 1818, which is a subjective synonym of *Cochlodesma praetenue* (Pulteney, 1799), an European anomalodesmatan species of the family Periplomatidae. However, Habe used it in the sense of *Exolaternula spengleri* (Gmelin, 1791); the name *Exolaternula* is thus based on a misidentified type species and a type species needs to be fixed under *ICZN (1999)* Art. 70.3. The available literature data on shell and anatomical characters of "*Anatina truncata*" or "*Laternula truncata*" (*e.g.*, *Ridewood, 1903*; *Burne, 1920*; *Morton, 1973*, *1976*; *Adal & Morton, 1973*; *Sartori, Passos & Domaneschi, 2006*) are referable to *E. spengleri* (of which *E. rostrata* (G.B. Sowerby II, 1839) is another synonym) and thus fall under the current concept of *Exolaternula*.

Other early anatomical studies have been variously interpreted as referring to species of either genus. *Woodward (1855*: 26) figured and described the anatomy of "*Anatina subrostrata*" from the Philippines, which is a synonym of *L. anatina* (the type species of *Laternula*). *Morton (1976*: 263) claimed that Woodward reported on "*L. rostrata* (= *L. truncata*)", a synonym of the type species of *Exolaternula*. However, *Exolaternula* species retain a lithodesma throughout their ontogeny and this structure is not represented in Woodward's figure. Considering the shell shape of the figured specimen and the reported locality (Philippines), it seems more likely that Woodward studied *L. corrugata*. *Pelseneer (1911*: 71–73, pl. 24) provided a detailed anatomy of *Anatina subrostrata*, which is a synonym of *L. anatina* (the type species of *Laternula*). However, *Morton (1976*: 263) stated this to be "(= *L. anserifera*)", which is a synonym of *Exolaternula spengleri* according to *Taylor et al. (2018)*. Other studied species have also been synonymized or reidentified, such as *L. marilina* Reeve (1860) (examined, *e.g.*, by *Sartori, Passos & Domaneschi (2006)* from Moreton Bay, Australia), now a synonym of *L. gracilis* (Reeve, 1860). The species recorded by *Prezant et al. (2008*, *2015)* as *L. corrugata* or *L. anatina* from Kungkraben Bay,

Thailand, has been recognized as a different species, *Laternula* sp., based on molecular analysis by *Taylor et al. (2018)*.

The deep-burrowing habit of *Laternula elliptica*, with highly extendable siphons, has been interpreted as allowing it to avoid predation and ice scouring (*e.g.*, *Ahn, 1994*; *Harper et al., 2012*).

The mode of operation of its valves and of other representatives of *Laternula* was described by *Morton (1976)* and *Savazzi (1990)*. *Morton (1976)* claimed that in *L. truncata* and *L. boschasina* the lithodesma immobilizes the ligament. *Sartori (2009)* observed that in several anomalodesmatans a lithodesma is formed by the calcification of the sagittal portion of the early juvenile ligament (ligament 1 or L1). In many species L1 is retained as the sole ligament throughout ontogeny but, in many others, including *L. elliptica*, a second ligament (L2) forms behind L1. As ontogeny progresses and L2 grows, in *L. elliptica* the lithodesma is gradually absorbed and L1 resilifers are overgrown. Hence, contrary to the observations made by *Peck et al. (2004*: 359), adult specimens of *L. elliptica* do not possess a lithodesma.

The siphons possess true tentacular eyes as in *E. spengleri* (*Morton, 1973*; *Adal & Morton, 1973*; as *L. truncata*), a possible adaptation to life in deep permanent burrows with little body movement, relying on siphonal retraction for defense. Also, arenophilic glands were described for the Laternulidae by *Sartori, Passos & Domaneschi (2006)*, who pointed out that, in this family, the glands are mostly restricted to the tip of the siphons. *Sartori, Passos & Domaneschi (2006)* further suggested the presence of arenophilic glands is a synapomorphy of the Anomalodesmata, and that in some of its families (Thraciidae, Cleidothaeriidae and Myochamidae) they have been lost. The presence of living hydrozoans, bryozoans and filamentous algae attached to the periostracum of the siphons suggests that these organs are not frequently disturbed.

In the adults of *L. elliptica*, a nonfunctional byssal groove was observed in the ventral part of the foot. The byssus likely is present in the larval stages of the species, and the byssal gland becomes reduced after metamorphosis. When the animal is displaced from its natural position in the substratum, the foot is used in burrowing, but this repositioning in the sediment takes hours, in contrast to the more rapid burying by juveniles, who possess a comparatively longer and more mobile distal portion of the foot. As discussed by *Morley et al. (2007b)*, *L. elliptica* has 25–30% longer relative foot length than tropical congeners of the same size, which could be a morphological adaptation compensating for reduced burrowing speeds in a colder environment.

*L. elliptica* may be regarded as a specialized detritus suspension feeder, collecting material in suspension near the sediment surface. Within the mantle cavity, the organs concerned with the collection, sorting and either acceptance or rejection of this material are well developed. The ctenidia are plicate, passing food material into the ventral marginal food groove of the inner demibranch only. The labial palps and the rejectory tracts of the mantle and visceral mass are efficient, this being probably related to a large amount of material that enters the mantle cavity.

*Sartori (2009)* examined the anatomy of numerous anomalodesmatans and noted that an inter-chamber aperture appears to be present in all members of the group bearing

ctenidia. In *L. elliptica*, this aperture plays a role in its burrowing process. To move deeper into the stiff, muddy substratum, completely buried individuals of *L. elliptica* profit from hydraulic burrowing mechanisms, powered by extra-water previously retained within the capacious lumina of both suprabranchial chamber and exhalant siphon. Forcibly transferred *via* the inter-chamber aperture onto the infrabranchial chamber, such extra water allows an extended jetting that lasts more than one would expect in a typical siphonate bivalve lacking such inter-chamber communication. The function of the cilia present along the free border of the inter-chamber valve and of the free, tentacle-like tips of the ctenidial axes still deserve investigation.

## CONCLUSIONS

Prior observations on the anatomy of *Laternula elliptica* were based on limited, preserved, and partly damaged material. The current work greatly expands on, and corrects, earlier observations. Among them were the initial reports by *Burne (1920)*, who missed anatomical features such as the presence of optical tentacles and interpreted a connection of the gill axis to the body wall by a "membranous sheet" (the latter likely was an artifact because of contortion of the single, damaged, specimen at his disposal; Burne's figure 20, plate IV). Among the noteworthy findings of the present study are the presence of well-developed siphons furnished with sensory tentacles at its tips, some of which bear eyes; large, folded gills and labial palps capable of sorting the material entering the mantle cavity; an inter-chamber communication in the posterior region of the mantle cavity; an ample ventral mantle fusion with an anterior pedal gape; the absence of a $4^{th}$ pallial opening; and the absence of a ligamental lithodesma in adult specimens. Benefiting from the careful dissections and live-animal observations during field studies conducted by the late Osmar Domaneschi, details could be explored that reveal the anatomical and behavioral features of this giant and important Antarctic keystone bivalve species.

## ACKNOWLEDGEMENTS

The field-observation-based project was originally conceptualized by the late Professor Walter Narchi and executed by the late Professor Osmar Domaneschi, who served as an excellent mentor for two of the current authors (F.D.P. and A.S.). We also acknowledge the divers Tânia Brito and Luciano Candisani from the Oceanographic Institute of the University of São Paulo (IOUSP) for underwater observations. We greatly appreciate the constructive input by Robert Prezant, John Taylor, and an anonymous colleague during the review process.

### Funding

This work was carried out within the Brazilian Antarctic Programme (PROANTAR), with financial and logistic support provided by the National Council of Scientific and Technological Development (CNPq), the Brazilian Navy and Brazilian Air Force. It was also supported by scholarships from São Paulo Research Foundation (FAPESP)

(Proc. 99/02399-9) and from "Pós-Graduação, Área Zoologia, IBUSP". FAPESP also provided financial support through grants no. 2018/06347-6 and 2018/10313-0. Relevant bivalve research at the Field Museum was supported by U.S. National Science Foundation (NSF) award DEB-0732854 to Rüdiger Bieler for the Bivalve-Tree-of-Life (BivAToL) project. The funders had no role in study design, data collection and analysis, decision to publish, or preparation of the manuscript.

## Grant Disclosures

The following grant information was disclosed by the authors:
Brazilian Antarctic Programme (PROANTAR).
National Council of Scientific and Technological Development (CNPq).
Brazilian Navy and Brazilian Air Force.
São Paulo Research Foundation (FAPESP): 99/02399-9.
Pós-Graduação, Área Zoologia, IBUSP.
FAPESP: 2018/06347-6 and 2018/10313-0.
U.S. National Science Foundation (NSF) Award: DEB-0732854.
Rüdiger Bieler for the Bivalve-Tree-of-Life (BivAToL) Project.

## Competing Interests

Rüdiger Bieler is an Academic Editor for PeerJ.

## Author Contributions

- Flávio Dias Passos performed the experiments, analyzed the data, prepared figures and/or tables, authored or reviewed drafts of the article, and approved the final draft.
- André Fernando Sartori performed the experiments, analyzed the data, authored or reviewed drafts of the article, and approved the final draft.
- Osmar Domaneschi conceived and designed the experiments, performed the experiments, analyzed the data, prepared figures and/or tables.
- Rüdiger Bieler performed the experiments, analyzed the data, authored or reviewed drafts of the article, and approved the final draft.

## Data Availability

This study is based on direct observations of morphological features. The findings are documented in the form of photographs and drawings (figures).

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
