# Peer review of "Anatomy and behavior of Laternula elliptica, a keystone species of the Antarctic benthos (Bivalvia: Anomalodesmata: Laternulidae)"

_PeerJ, doi:10.7717/peerj.14380_

## Round 0.1 · original submission · Minor Revisions

The three reviewers unanimously underline the quality of the manuscript, but have identified a number of minor issues that should be resolved before final acceptance. I ask that, among other things, the opening statement of the title ("Secrets of a giant ...") be reconsidered. Please also note the three annotated manuscripts, which contain further detailed suggestions.

·

Basic reporting

This is a clear and well written manuscript that contains relevant information to flesh out some of the anatomical and behavioral information on an important Antarctic species of bivalve. The manuscript attends to the abundant literature on this taxon offering a set of previously unknown or poorly known structural/anatomical features. The behavioral information, mainly on burrowing and palp/ctenidial ciliary currents, represents notes and diagrams from unpublished observations from the 1990s by Osmar Domaneschi (deceased) and entrusted to his students.

The anatomical reporting, while not of the detail of some focused functional morphology papers on the bivalves, is meant to highlight the information the authors perceive as missing and important to our better understanding this “keystone” species. The authors note some structures as “noteworthy”, and while not necessarily unique in form, this is a fair descriptor as the data fill gaps.

The manuscript structure conforms to standards and the figures are of good quality. Additional support could be supplied by some histological micrographs that would help the reader visualize some of their cell/tissue observations.

Experimental design

The behavior observations are just that, albeit they are translated from the information handed down from Domaneschi. The other techniques used are (histology, SEM) standard practice. Some clarification of the fixation methods used for histology and SEM would be helpful as the specimens were collected in the 1990s and the citations for preparation are from the 2000s.

Validity of the findings

As the behavioral observations are historical (1990s), replication adjacent to the time frame are, of course, not possible. With the relative difficulty of obtaining and maintaining live specimens and the uniqueness of the observations, the descriptive information added here is relevant to better understanding laternulid bivalves in general. The anatomical information Domaneschi uncovered from dissections of living animals is important and, along with his behavioral observations, helps us better appreciate the functional morphology of L. elliptica. While the gross morphological structures discussed are not unusual for many members of the Anomalodesmata, the data shared here are helpful in creating a more holistic picture of the details (as per authors, the “secrets”) of this species.

Additional comments

I enjoyed reading this paper and offer comments and questions within the text that are self-explanatory. In particular:
1. clarify statements regarding the relationship between external particle adhesion, the arenophilic glands vs adhesion to the periostracum. (lines 141, 219-224) - is periostracum the adhesive surface in some areas on the shell vs secretions from arenophilic glands offering the adhesion for other areas (specifically siphonal)?
2. Consider the addition of some histological photographs to better document some of the descriptions (and to support the line drawn figures).
3. The 4th pallial aperture is found in some anomalodesmatans, other bivalves, with various functions assigned. Was such an opening found in L. elliptica? (190) The openings can be small and easily missed. The burrowing behavior revealed in this paper would align with some of the functionality given to this extra aperture in bivalves with relatively fused mantle margins.
4. Clarify some of the palp behaviors observed in terms of the possibility that the activities of the palp in a “bivalve on the half shell” may not align with a fully adducted clam (secured by both valves and covering mantle tissue).
5. The lithodesma is an important functional and taxonomic character of some anomalodesmatans. A bit more discussion would be of value. Care should also be taken in how the lithodesma is described in terms of being a “ligament” (487-488) – when present the lithodesma may be an adjunct of the ligament but a fully calcified lithodesma does not have the elasticity to serve as a ligament.
6. Were the gut contents examined, thus allowing confirmation of the statement that this species is a "specialized detritus suspension feeder"? Is this supported by other literature? If so, that needs to be cited. (510).
7. The additional taxonomic discussion and clarification is much welcome. The confusion in taxonomy of this group reflects and supports the need for this paper in helping us distinguish and better understand the relatively few species in this family.

Reviewer 2 ·

Basic reporting

No comment

Experimental design

No comment

Validity of the findings

No comment

Additional comments

This manuscript is a detailed and straightforward descriptive account of the mantle organs and burial behaviour of a well-known anomalodesmatan bivalve in the Southern Ocean. The manuscript is well-organised with relevant and clear illustrations. I have no major criticisms. However, given the level of detail provided for the sorting and ciliary mechanisms observed in the ctenidia and labial palps, I was a little surprised that there is no mention about the rest of the internal digestive system within the animal (in particular, the stomach, intestine and rectum). Although the authors have qualified their study as having a special focus on pallial organs, it is also stated that ‘this paper reviews the comparative anatomy, functional morphology…of L. elliptica…’ (lines 94-95). I feel it would be helpful to complete the picture by providing brief descriptions of the remainder of the gut as well as the positions of other key organs visible from the surface (e.g., heart, gonads), even if they can only be based on published material (e.g., Burne, 1920; Morton, 1973).

In the Discussion section, I was also looking forward to reading about the secrets that the authors promise to reveal, but I was a little disappointed that none were highlighted. Instead, the section begins with three paragraphs centred on a taxonomic review of laternulid species. The relevance of this review seems unclear, without a discussion of the function(s) of the lithodesma first (i.e., the 3rd paragraph). Perhaps a re-organisation of the Discussion section to emphasise the significance of the findings, preferably in order of their importance, would do better justice to the given title.

Other comments and suggested corrections are provided directly in the attached pdf.

Annotated reviews are not available for download in order to protect the identity of reviewers who chose to remain anonymous.

·

Basic reporting

On the whole well organised. Some small issues with English. Some parts of the text difficult to follow without better figures.

Previous literature and background well documented - perhaps too much (see comment on pdf) . Need to check citations of Ansell & Rhodes 1997 against Ansell & Harvey 1997

The authors are correct in stating that Laternula elliptica is an important species of benthic communities around Antarctica. There have been many studies of its biology and ecology but surprisingly the basic anatomy is poorly documented.

The authors attempt to redress this lack with observations on live animals and an account of the general anatomy. This is fine as far as it goes but they only consider the organs of the pallial cavity and siphons. For example, there is no description of the excretory or reproductive systems. They state that histological thin sections were made but we only have details of the siphon musculature. Although there is some difficult to follow discussion in the text about mantle fusion there are no diagrams of this or images of histological sections of mantle edges. Similarly, there are no detailed images of gill structure except for a poor SEM figure. Also, what about presence of a lithodesma in juvenile L. elliptica -mentioned but not followed up.

The observations on burrowing behaviour and surface siphoning activity have been described before by several authors.

As far as it goes there is some useful information here that would allow comparison with temperate and tropical Laternula species. But it could have been so much better. Up to date methods using CT scans, and more detailed histology could have taken it to a different level.

Many comments inserted on the pdf.

Experimental design

Field and lab observations plus histology

Validity of the findings

Observations generally useful but could have done much more

---

## Round 0.2 · accepted · Accept

Thank you very much for the thorough revision which has resolved all pending issues. Please correct the two minor points identified by reviewer 3 (John Taylor) in lines 465, 469 and 547. With these corrections, your manuscript is ready for publication.

·

Basic reporting

Issues resolved from first copy.

Experimental design

Issues resolved from first copy.

Validity of the findings

Issues resolved from first copy.

·

Basic reporting

The authors have attended to most of the points raised by the reviewers and their suggestions for improving the manuscript. New text has been added to cover excretory and reproductive characters and the descriptions of the ctenidia and mantle fusion are much improved. Additions have been made to some of the figures that have improved documentation of the anatomical features.

The whole text is much improved and I have no further issues.

Two small points

line 547 spelling of Morley

lines 465, 469. I would prefer use of testis, testes rather than testicle. The latter more usually associated with external testes as in vertebrates.

Experimental design

no comment

Validity of the findings

no comment